# DNA Methylation and RNA-Sequencing Analysis to Identify Genes Related to Spontaneous Leaf Spots in a Wheat Variety 'Zhongkenuomai No.1'

**Zhibin Xu** [1,†]**, Fang Wang** [2,†]**, Xiaoli Fan** [1]**, Bo Feng** [1]**, Qiang Zhou** [1]**, Qichang Yang** [2,*] **and Tao Wang** [1,3,*]

1  Chengdu Institute of Biology, Chinese Academy of Sciences, Chengdu 610041, China; xuzb@cib.ac.cn (Z.X.); fanxl@cib.ac.cn (X.F.); fengbo@cib.ac.cn (B.F.); zhouqiang@cib.ac.cn (Q.Z.)
2  Institute of Urban Agriculture, Chinese Academy of Agricultural Sciences, Chengdu 610213, China; nice_fangwang@163.com
3  The Innovative Academy of Seed Design, Chinese Academy of Sciences, Beijing 100101, China
*  Correspondence: yangqichang@caas.cn (Q.Y.); wangtao@cib.ac.cn (T.W.)
†  These authors contributed equally to this work.

**Abstract:** Greenish leaf variation has been reported widely as a trait of great interest in wheat for improving photosynthesis. Zhongkenuomai No.1 (ZKNM1), a mutant with recoverable leaf spots, was regarded as a suitable material for studying chlorophyll synthesis-related mechanisms. In this study, transcriptome and DNA methylation analyses were conducted in ZKNM1 leaves to determine the transcriptional regulatory mechanism of leaf spot development. Ultimately, 890 differentially expressed genes (DEGs) were discovered, with chlorophyll biosynthesis pathway genes downregulated and chlorophyll degradation pathway genes upregulated, possibly acting as a double block to chlorophyll accumulation. Among them, *HEMA1s (Glutamyl-tRNA reductase family proteins)* and *PORAs (protochlorophyllide oxidoreductase A)* were the most important controlled genes. Furthermore, a genome-wide methylation analysis indicates that a hypermethylated region is present 1690 bp upstream of the transcriptional start sites in spot tissues (SPs), and 131 DNA methylation-mediated DEGs were identified, one of which encoded a putative resistance gene (*TraesCS1A02G009500*) and was a hub gene in interaction network modules. In the sample groups with leaf spots (SPs), this gene may be involved in the photosynthetic processes. The findings indicated that dynamic variations in DNA methylation play key roles in gene regulation to govern leaf spot development.

**Keywords:** DNA methylation; transcriptome; leaf color; wheat; chlorophyll metabolism

## 1. Introduction

Wheat (*Triticum aestivum*) is an essential food crop, accounting for approximately one-fifth of the calories and protein consumed by mankind [1]. With the projected rise in population and food demand, global production of grain crops such as wheat will need to increase by more than 60% by 2050 [2,3].

Leaf color variation is an important mutation for improving photosynthesis and increasing yield potential. In fact, several stably inheritable leaf color variations have been discovered in wheat (*Dlm*, *lm3*, *Ygm* and *chli*), providing abundant genetic resources for breeding wheat with high photosynthetic efficiency and ideal yield potential [4–7]. Most leaf color mutants can be primarily divided into two types: homogeneous (just one leaf color: albina, xantha mutations) and heterogeneous (two or more leaf colors: alboviridis, viridis, or tigrina mutations). These leaf color variations are usually associated with the abnormal metabolism of plant pigments, including chlorophyll, carotenoids, and anthocyanins. Most mutant genes can either directly or indirectly affect chlorophyll production, degradation, content, and proportion, which can affect photosynthesis [8] and are therefore extensively studied [9].

Chlorophyll biosynthesis is a complex process involving fifteen steps and approximately twenty-seven genes encoding fifteen enzymes [8]. In general, the first step of chlorophyll biosynthesis is catalyzed by glutamyl-tRNA reductase (GluTR), which is encoded by *Glutamyl-tRNA reductase 1* (*HEMA*) genes and reduces the activated glutamate to glutamate-1-semialdehyde (GSA). This step is also characterized as the first committed step in the synthesis of 5-aminolevulinic acid (ALA), a precursor of chlorophyll and the primary light-harvesting pigment of higher plants and is thus thought to be a crucial control point in regulating chlorophyll availability [10,11]. In *Arabidopsis*, two *HEMA* genes exist. *HEMA1* is the dominant form in leaves and is expressed in all parts of plants [12], whereas *HEMA2* was found only in roots and flowers [13]. When the expression of *HEMA1* was inhibited, plants exhibited varying degrees of chlorophyll deficiency, ranging from showing a patchy area of yellow to being total completely yellow [10]. Another critical step in chlorophyll synthesis is the reduction of protochlorophyllide (Pchlide) to chlorophyllide (Chlide), which is catalyzed by protochlorophyllide oxidoreductases (PORs) encoded by POR genes [14]. Different POR isoforms are found in different plants and exhibit distinct expression patterns [15]. In barley, two PORs were found: *PORA* and *PORB*. *HvPORA* is primarily expressed during etiolation, germination, and greening, and declines during illumination [16]. Collectively, blocking in the chlorophyll biosynthesis process may result in leaf color mutations. The process of leaf color fluctuation is highly complex and warrants additional investigation in various plants.

Most wheat leaf color mutants, particularly chlorophyll-deficient mutants, present decreased production performance [17]. Compared to the non-recoverable degreening leaf color variation, staged, the regreening leaf color variation has relatively little influence on the overall accumulation of photosynthetic assimilates. Furthermore, displaying distinct leaf colors at different growth phases would result in a more diversified and perceptible leaf landscape. Wang et al. reported a pale-green leaf mutant (*chli*) presenting greenish leaves from the one-leaf stage and re-green at the heading stage [7]. The results revealed no change in kernel quantity or seed setting rate with the wide type, but a notable reduction in tiller number. Most leaf color mutations, such as *chli*, are generally expressed at the seedling stage [8]. Therefore, dissecting the molecular basis involving leaf color mutation would provide more as viable targets for genetic improvement.

DNA methylation is an important and conserved epigenetic modification [18] that happens in CG, and CHG contexts (symmetrical), or in CHH contexts (asymmetrical) [19]. DNA methylation has been shown to influence a variety of leaf features, including growth [20], senescence [21], shape [22], and pigmentation [23,24]. Among them, Cocciolone and Cone found that the hypermethylation of a maize anthocyanin regulatory gene (*P1-Bh*) might lead to the linear distribution of pigmented cells and thus the leaf presented variegated stripes [23]. Wang et al. reported that CG hypermethylation might be associated with sectorial decreases in chlorophyll levels and result in the formation of *Cmvv* striped leaves [24]. However, it remains unclear how DNA methylation influences the formation of leaf spots in wheat.

In this study, a wheat variety, Zhongkenuomai No.1 (ZKNM1), with a spontaneous and recoverable variation in leaf color at the adult plant stage was reported. ZKNM1 was released in 2015 by the Chengdu Institute of Biology, CAS. It is waxy wheat with hereditary nonpathological greenish leaf spots (Figure 1) and good yield performance. These greenish spots usually appear around the flag leaf emergence stage and regreen at the late filling stage. These recoverable leaf spots can potentially present a dynamic landscape with minor yield limitations. However, the mechanism causing this trait is still unknown. In this study, the transcriptome and DNA methylation investigation on ZKNM1 identifies the related pathways or genes associated with the leaf spot dynamics and uses them to facilitate future high photosynthetic-efficiency wheat breeding.

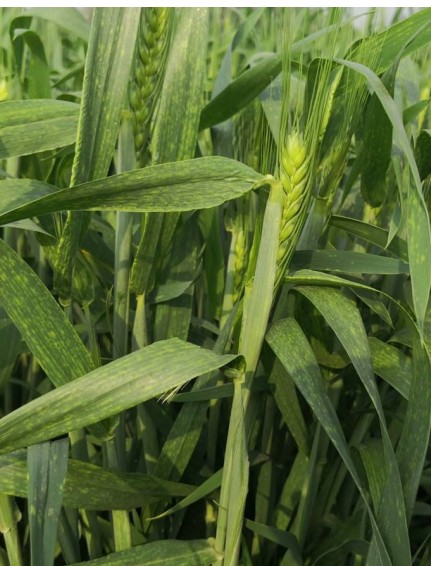

**Figure 1.** The leaf spots of Zhongkenuomai No.1 in the field environment.

## 2. Materials and Methods

### 2.1. Sample Collection

In this study, two types of ZKNM1 flag leaves (leaf tissues without spots (CK) and tissues with spots (SP)) were sampled at the heading stage. Three replicates were performed in this study. Flag leaf tissues from three independent plants were excised using a hole punch and pooled together for each replicate. Each replicate was crushed together and separated for DNA and RNA to ensure they were pooled from the same sample.

### 2.2. Transcriptome Sequencing and Data Analysis

Total RNA was extracted from the sampled leaves with the Plant RNA Kit (Omega, Norcross, GA, USA) and then treated with DNase I (Takara, Dalian, China). The total RNA quality and quantity were analyzed by the Bioanalyzer 2100 and RNA 6000 Nano Lab Chip Kit (Agilent, Palo Alto, CA, USA) with RIN number >7.0 [4]. Both cDNA library preparation and transcriptome sequencing were conducted by Biomarker Technology Co. (Beijing, China). All 6 libraries were sequenced using the Illumina NovaSeq 6000 platform (San Diego, CA, USA). Raw data (raw reads) were processed using NGS QC Toolkit. The reads containing ploy-N and the low-quality reads were removed. At the same time, Q30 was calculated. Clean RNA-seq data were obtained and bioinformatics analysis was performed using BMKCloud platform (www.biocloud.net, accessed on 1 June 2022). The clean reads were mapped to the International Wheat Genome Sequencing Consortium wheat reference genome (IWGSC RefSeq 2.0) (https://urgi.versailles.inrae.fr/blast_iwgsc/?dbgroup=wheat_iwgsc_refseq_v2_chromosomes&program=blastn, accessed on 1 June 2022) using HISAT 2.0 [25]. Only uniquely mapped reads were analyzed further and annotated based on the reference genome.

### 2.3. Differentially Expressed Gene Analysis

FPKM was used as the unit of measurement to estimate transcript abundance. The differentially expressed genes (DEGs) were selected using DESeq2 [26] on the BMKCloud online platform (http://www.biocloud.net/, accessed on 1 June 2022). DEGs were identified using the DESeq functions estimateSizeFactors and nbinomTest. The resulting $p$ value was adjusted using Benjamini and Hochberg's approach for controlling the false discovery rate (FDR). Adjusted $p$ value ($q$ value, FDR) <0.05 and $|\log_2$ fold change$| > 1$ was set as the threshold for significant differential expression. Hierarchical cluster analysis of DEGs was performed to explore transcript expression patterns.

### 2.4. Gene Functional Annotation

The discovery of novel genes was achieved by StringTie [27] compared with the annotated IWGSC RefSeq v2.0. The short transcripts (coding peptides <50 amino acids) were excluded. The obtained transcript regions without annotation were defined as novel genes.

Gene function was annotated based on the following databases: Gene Ontology (GO) [28], Kyoto Encyclopedia of Genes and Genomes (KEGG) [29], Clusters of Orthologous Groups of proteins (COG) [30], evolutionary genealogy of genes: Non-supervised Orthologous Groups (eggNOG) [31], euKaryotic Orthologous Groups (KOG) [32], Protein family (Pfam) [33], Swiss-Prot [34], and NCBI non-redundant (Nr) protein sequences [27]. The transcription factors (TF) were predicted based on plantTFDB database. Gene functional annotation and TF prediction were performed on the BMKCloud online platform (http://www.biocloud.net/, accessed on 1 June 2022). Among these, GO enrichment analysis of differentially expressed genes was implemented by the clusterProfiler R package, in which gene length bias was corrected. GO terms with adjusted $p$ value ($q$ value, FDR) < 0.05 were considered significantly enriched by differential expressed genes. The clusterProfiler R package was used to test the statistical enrichment of differential expression genes in KEGG pathways.

### 2.5. Gene Set Enrichment Analysis

All expressed genes were subjected to gene set enrichment analysis (GSEA) [35] on the online platform OmicShare 6.3.9 (https://www.omicshare.com/, accessed on 1 June 2022) to identify whether a set of genes in specific GO terms or KEGG pathways showed significant differences between the CK and SP groups. Briefly, we inputted the gene expression matrix and ranked genes by SignalToNoise normalization method. Enrichment scores and $p$-value were calculated. The significant enrichment of terms and pathways was selected based on the absolute values of enrichment score (ES) >0.3, nominal $p$-value < 0.05, normalized enriched score (NES) >1, gene size >15 and <500, and false discovery rate (FDR) <0.01.

### 2.6. Protein–Protein Interaction (PPI) Network Construction and Modules Selection

The Search Tool for the Retrieval of Interacting Genes (STRING) database (http://string-db.org/, accessed on 23 June 2022) [36] was used to obtain the predicted protein–protein interaction (PPI) relationship of these DEGs. Then the PPI of these DEGs was visualized in Cytoscape (http://cytoscape.org/, accessed on 1 June 2022) [37]. The CytoHubba application in Cytoscape was performed to analyze the hub genes through five centrality methods, including Degree, EPC (Edge Percolated Component), EcCentricity, Stress, and MCC (Maximal Clique Centrality) [38]. The Molecular Complex Detection (MCODE) [39] application in Cytoscape was used to screen the hub modules of the PPI network. The criteria setting of MCODE was degree cutoff = 2, node score cutoff = 0.2, k-core = 2, and maximum depth = 100.

### 2.7. MethylRAD Sequencing

Genomic DNA of the sampled flag leaves was extracted using the phenol–chloroform extraction method to construct the MethylRAD library and sequencing was performed by OE Biotech Co., Ltd. (Shanghai, China). Six libraries were constructed following the protocol from Wang et al. using the Illumina Nova PE150PE platform [40].

### 2.8. MethylRAD Data Analysis

Raw reads were first subjected to quality filtering and adaptor trimming. Reads with ambiguous bases (N) exceeding 8%, poor quality (15% nucleotide positions with a Phred quality < 30), or without enzyme restriction sites were removed by a custom script in Perl. Then, the reads with enzyme sites were subsequently aligned against the genome of IWGSC RefSeq v2.0 using Bowtie 2 (version 2.3.4.1) [41]. Based on the location information of the methylation sites, the sites were annotated using SnpEff software (version 4.1g), giving the

gene element where each site is located and the detailed information of the site annotation, which contains the gene element, gene name, gene id, and transcript id. The distribution of methylation sites in different gene elements in each sample was counted. The relative expression level of each methylated site (CpG) was determined using the normalized read depth RPM (reads per million). The 2 kb upstream and downstream regions of the transcription start site (TSS), the 2 kb upstream and downstream regions of the transcription stop site (TTS), and the genetic region (genebody) were selected, and the distribution trend of sequenced reads in these regions was calculated. The change in methylation level was assessed based on the sequencing depth of each site in the relative quantitative results of methylation, using R package DESeq [42]. The differentially methylated sites (DMSs) were screened with the criteria of $|\log_2$ fold change$| >1$ and $p$ value $< 0.05$. The DMSs located in the upstream region denoted differentially methylated promoter (DMP) regions. The genes assigned with DMSs were used for functional annotation and integrated analysis with DEGs.

### 2.9. GO, KEGG Enrichment, and Integrated Analysis

To identify the biological functions related to the DEGs and DMSs, the enrichment analysis of KEGG pathways and GO terms (CC, Cellular Component, MF, Molecular Function, BP, Biological Process) was implemented and visualized. The $p$ value $< 0.05$ was set as the threshold for both GO and KEGG enrichment. Terms and pathways meeting this condition were defined as significantly enriched GO terms and KEGG pathways. Four-quadrant analysis between DMSs and DEGs was conducted to assess the potential correlation between DNA methylation and gene expression. The above analyses were conducted on the online platform OmicShare 6.3.9 (https://www.omicshare.com/, accessed on 1 June 2022).

### 2.10. qPCR Analysis

Leaf sample RNA was extracted using TRIzol reagent (Invitrogen, Carlsbad, CA, USA) and reverse transcribed into cDNA, which was used as a template for qPCR. qPCR validation for 10 genes was performed using ChamQ Universal SYBR qPCR Master Mix (Vazyme, Nanjing, China). The primers are listed in Table S1. The qPCR samples were the same as the samples utilized for RNA-seq. All the reactions were performed in triplicate for each sample.

### 2.11. Yeast Two-Hybrid (Y2H)

The full-length CDS of *TaXA21 (TraesCS1A02G009500)* and *TaHY5* were cloned into the pGBKT7 and pGADT7 vectors, respectively. The bait and prey constructs were co-transformed into yeast strain Y2H using the lithium acetate method. Then, transformants were selected on growth media (SD/-Leu/-Trp, SD/-Ade/-His/-Leu/-Trp, SD/-Ade/-His/-Leu/-Trp/X-α-Gal), and the positive interaction transformants were cultivated into a medium for serial dilution. Exponentially grown yeast cells were harvested and adjusted to an optical density at 600 nm (OD600) of 0.5 with sterilized double-distilled water. Then, the cell suspension was diluted 1/10, 1/100, and 1/1000. Dilutions (5 μL) of the yeast cells were spotted onto SD/-Leu/-Trp, SD/-Ade/-His/-Leu/-Trp media, grown at 30 °C for 3 days, and photographed.

## 3. Results

### 3.1. Transcriptome Assembly and Functional Annotation

RNA sequencing was used to discover differentially expressed genes probably related to leaf spot development in two leaf tissue groups with or without spots (spotted tissue, SP, and normal tissue, CK group). Six cDNA libraries were produced in total, yielding 37.79 Gb of clean data, with 5.75–6.89 Gb of clean data obtained for each sample, and the Q30 base distribution varied from 91.79 to 92.31% (Table S2). The raw reads were uploaded to NCBI (SRA accession no. PRJNA827248). After mapping to the reference wheat genome (IWGSC RefSeq 2.0), 92.17–93.24% of reads were mapped, with 86.81–88.07% uniquely mapped.

In total, 22,779 new genes were detected, 9150 of which were functionally annotated and characterized using the GOG/GO/KEGG/KOG/Pfam/Swiss-prot/TrEMBL/eggnog/Nr databases (Table S3; Additional file 1). A total of 6371 transcription factors (TFs) and 5374 protein kinase (PKs) were predicted (Additional file 2) based on the PlantTFDBe. The correlations among replicates for the two groups showed a modest variance and distinct variations between groups (Figure S1A), consistent with the result of principal component analysis (PCA) (Figure S1B). The first principal component, which accounted for 86.40% of the variance, SP, was significantly distinct from CK, indicating that the gene expression profiles were distinctive across the two tissues.

### 3.2. GSEA

GSEA was carried out using a GO-based list including 2731 gene sets and a KEGG-based list containing 120 gene sets (Additional file 3). Positive and negative NESs indicate higher and lower expression in the SP group, respectively. In total, 85 gene sets, including 70 GO-based gene sets and 15 KEGG-based gene sets, were identified as significantly enriched (FDR < 0.01). Among them, from the GO-based list, 44 lower and 26 higher expression gene sets in SP group were identified, and 5 lower and 10 higher expression gene sets in the SP group from the KEGG-based list are shown (Figure 2; Table S4). From the GO-based list, interestingly, the notably downregulated gene sets in the SP group, such as "photosynthesis, light harvesting" (Biological Process), "photosystem I" (Cellular Component), and "chlorophyll binding" (Molecular Function), were primarily related to photosynthesis and chlorophyll, and the upregulated gene sets were mainly involved in nicotianamine-related terms, such as "nicotianamine synthase activity" (Molecular Function) and "nicotianamine metabolic process" (Biological Process). From the KEGG-based list, the lower and higher expression gene sets in the SP group were "photosynthesis-antenna proteins" and "fatty acid degradation", respectively.

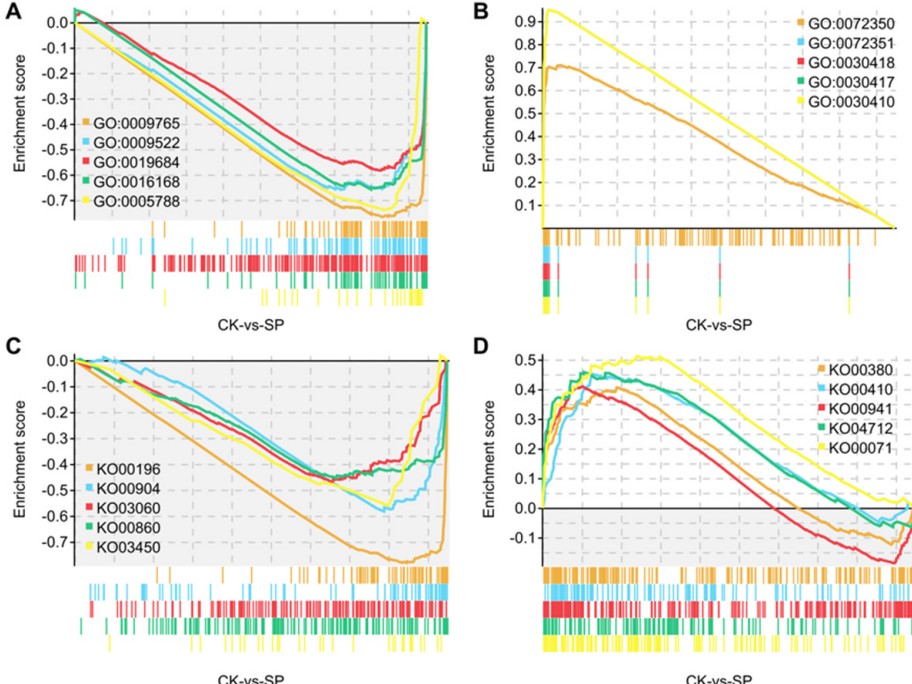

**Figure 2.** Gene set enrichment analysis (GSEA). Note: GSEA (Gene set enrichment analysis) was performed in the CK vs. SP comparison. ES (enrichment score) reflects the degree of overrepresentation at the top or bottom of the ranked list of the genes included in a gene set in a ranked list of all genes present in the RNA-seq dataset. The significantly enriched GO terms with the five lowest and highest NES (normalized enriched score) values (Table S4) were shown in (**A**,**B**), respectively. That is, (**A**): GO:0009765: photosynthesis, light harvesting; GO:0009522: photosystem I; GO:0019684:

photosynthesis, light reaction; GO:0016168: chlorophyll binding; GO:0005788: endoplasmic reticulum lumen; (**B**): GO:0030410: nicotianamine synthase activity; GO:0030417: nicotianamine metabolic process; GO:0030418: nicotianamine biosynthetic process; GO:0072351: tricarboxylic acid biosynthetic process; GO:0072350: tricarboxylic acid metabolic process. The significantly enriched KEGG pathways with the five lowest and highest values of NES (Table S4) were shown in (**C**,**D**), respectively. That is, (**C**): KO00196: Photosynthesis-antenna proteins; KO00904: Diterpenoid biosynthesis; KO03060: Protein export; KO00860: Porphyrin and chlorophyll metabolism; KO03450: Non-homologous end-joining; (**D**): KO00071: Fatty acid degradation; KO04712: Circadian rhythm-plant; KO00941: Flavonoid biosynthesis; KO00410: beta-Alanine metabolism; KO00380: Tryptophan metabolism.

### 3.3. Differentially Expressed Genes and Functional Analysis

As a result, CK and SP generated 889 DEGs, 428 of which were upregulated and 461 of which were downregulated, respectively (Figure S2A; Additional file 4). GO and KEGG enrichment analyses were performed to further understand the function of these DEGs. According to the GO enrichment analysis results, 692 DEGs were significantly enriched in 54 GO terms (level 1) (Figure S2B). DEGs associated with "metabolic process", "cellular process", and "single-organism process" were primarily enriched in the biological processes category. The GO enrichment (level 2) was further presented using directed acyclic graphs (DAG) (Figure S3), and the most highly enriched GO terms were related to chlorophyll, photosynthesis, or photomorphogenesis. In the biological processes category, for example, the GO terms "photosynthesis, light harvesting", "protein-chromophore linkage", and "nicotianamine biosynthetic process" were considerably enriched. Moreover, the GO terms "chlorophyll binding" and "nicotianamine synthase activity" for molecular functions, as well as the GO terms "photosystem I", "photosystem II", and "chloroplast thylakoid membrane" for cellular components, were highlighted.

According to KEGG enrichment analysis, 625 DEGs were enriched in 88 pathways (Figure 3; Additional file 4). Genes relevant to photosynthesis or chlorophyll biosynthesis-related pathways, such as "photosynthesis-antenna proteins", "alanine, aspartate, and glutamate metabolism", and "porphyrin and chlorophyll metabolism" were dramatically downregulated in the SP group, and "lysine degradation" genes were upregulated. However, the expression of genes related to "phenylpropanoid biosynthesis", "starch and sucrose metabolism", and "MAPK signaling pathway-plant" did not show obvious trends, showing partial upregulation and downregulation.

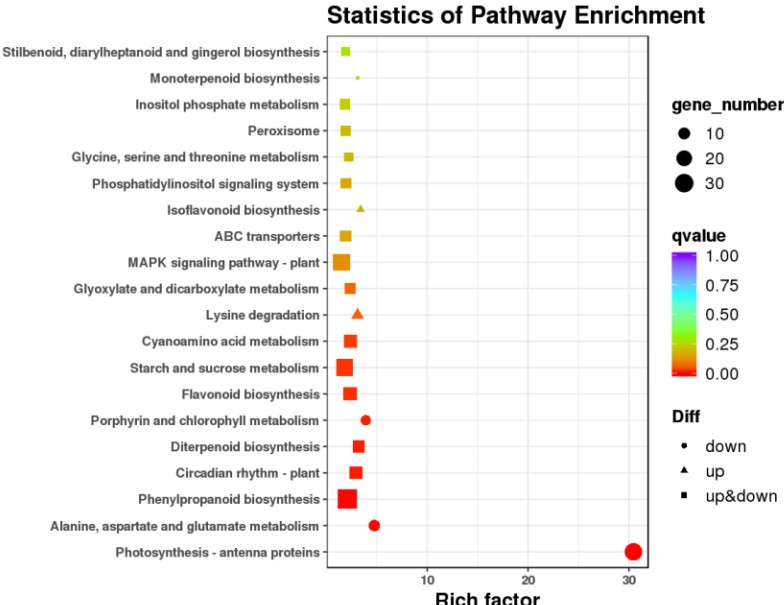

**Figure 3.** KEGG enrichment of DEGs. Note: The *y*-axis on the left represents KEGG pathways, and the *x*-axis indicates the "Enrichment factor". Enrichment factor is calculated as "Enrichment factor

= (Ratio of DEGs annotated to the term over all DEGs)/(Ratio of genes annotated to the term over all genes)". A larger enrichment factor indicates a more significant enrichment of the pathway. The dots indicated the down-expressed DEGs were enriched in this pathway; the triangles indicated the up-expressed DEGs were enriched in this pathway; and the triangles indicated both the down- and up-expressed DEGs were enriched in this pathway. The color of the dots/triangles/squares stand for adjusted $p$ value ($q$ value, FDR). The smaller the adjusted $p$ value ($q$ value, FDR) is, the more significant or reliable the enrichment is. The size of the dots/circles/squares represents the number of DEGs enriched in this pathway. The larger the dot is, the more genes it contains. KEGG: Kyoto Encyclopedia of Genes and Genome; DEGs: Differentially Expressed Genes.

The qPCR analysis showed that the RNA-seq data were well correlated with the qPCR results (Figure S4), with a relatively higher correlation coefficient $R^2$ (0.7816), indicating the reliability of the RNA-seq data.

### 3.4. Pigmentation Pathways

Because the functional annotation revealed chlorophyll pathways, the gene expression patterns of important chlorophyll metabolism genes were investigated further (Figure 4). Almost all genes (nearly 83.33%) involved in chlorophyll biosynthesis (such as HEMAs, GSAs, HEMBs, HEMCs, HEMHs, PPOX1s, GUN4s, CHLDs, CHLGs, CHLHs, CHLIs, CHLMs, CHLPs, CRD1s, DVRs, PORAs, and PORBs) showed decreased expression in the SP group as compared to that in the CK group. Significantly, both HEMA1s and PORAs, the relative upstream and downstream genes in the chlorophyll biosynthesis pathways, respectively, were significantly downregulated. In the SP group, however, most genes (almost 74.07 percent) that positively regulate chlorophyll degradation, such as NYC1s, NOLs, HCARs, PPHs, PAOs, and RCCR1s, were marginally upregulated.

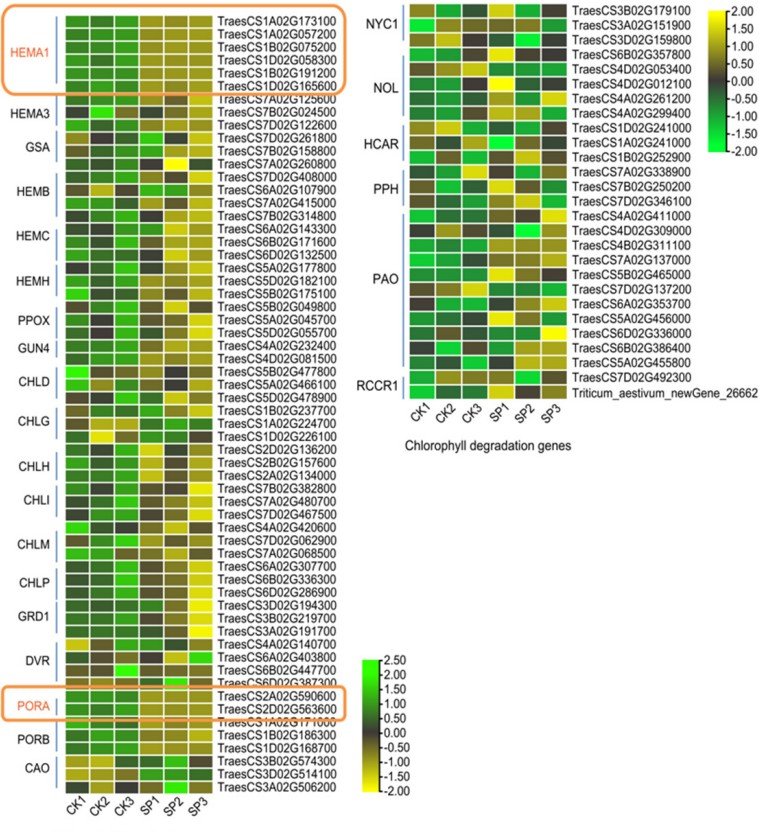

**Figure 4.** Expression profiling of genes involved in chlorophyll metabolism. Note: The relative expression values range from −2 to 2.

For the genes involved in carotenoid biosynthesis (DXRs, GGPSs, PSYs, PDS1s, ZDS1s, CRTISOs, CYP97s, ZEPs, and NCEDs), 23 and 31 of them were up- and downregulated in the SP group, respectively (Figure S5A). Furthermore, 28 and 34 of them affect anthocyanin biosynthesis (PALs, 4CLs, CHSs, CHIs, ANRs, UGT75L6s, 5MAT1s, and ANSs) were up and down-expressed genes in the SP group, respectively (Figure S5B).

### 3.5. Identification of Hub Genes through PPI Network Analysis of DEGs

The PPI network of DEGs, which had 147 nodes and 1389 edges, is presented in Figure 5. Table S5 lists the top 10% of the most significant DEGs (15 DEGs screened here) in the PPI network as determined by five centrality approaches from cytoHubba. All five approaches identified four common hub DEGs: Triticum aestivum newGene 7669, Triticum aestivum newGene 26895, TraesCS1A02G009500, and TraesCS2A02G559400 (Figure 6A; Table 1). Among these, Triticum aestivum newGene 7669, which encodes a bZIP type transcription factor, was found to be homologous to a tomato HY5 gene annotated by the SWISS database. MCODE built six major hub modules from the whole DEG PPI network, using the parameters of degree cutoff = 2, node score cutoff = 0.2, k-core = 2, and maximum depth = 100 (Figure S6). Among them, only module 5 (Figure 6B) comprised three hub genes. TraesCS1A02G009500, a homolog of rice Xa21, was designated as the seed gene.

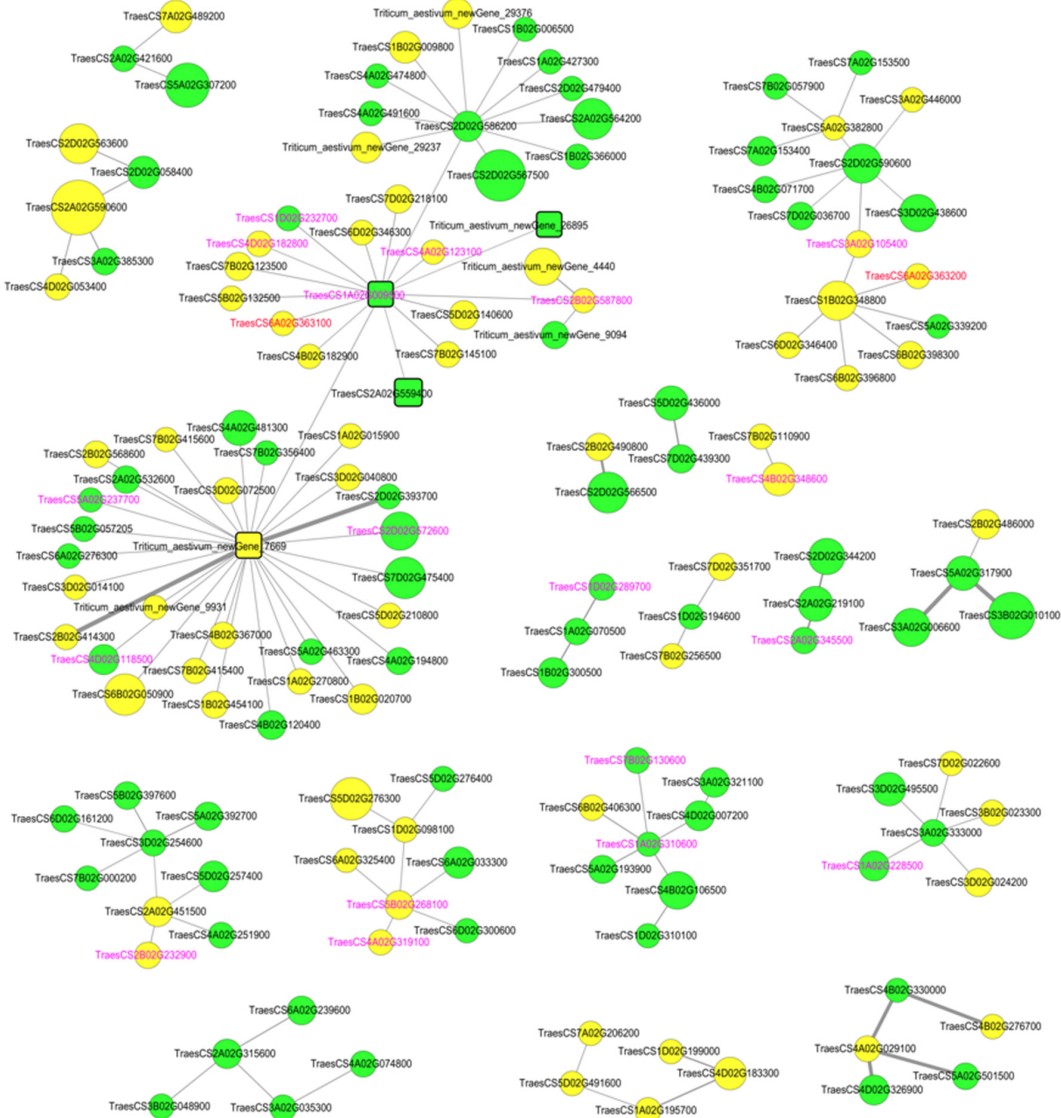

**Figure 5.** PPI network for DEGs. Note: The circles and rectangles indicate the DEGs and the hub

DEGs, respectively, detected in this study, and their size represents the fold change value of these DEGs. The thickness of lines between DEGs indicated the strengthening of their correlation. Green indicates the upregulated DEGs, and yellow indicates downregulated DEGs. The gene name in purple and red indicates these DEG are regulated by methylation, and the gene name in red indicates that the methylation levels in the promoter regions of the corresponding DEGs were significantly different between the CK and SP groups. PPI, Protein–Protein Interaction; DEGs, Differentially Expressed Genes.

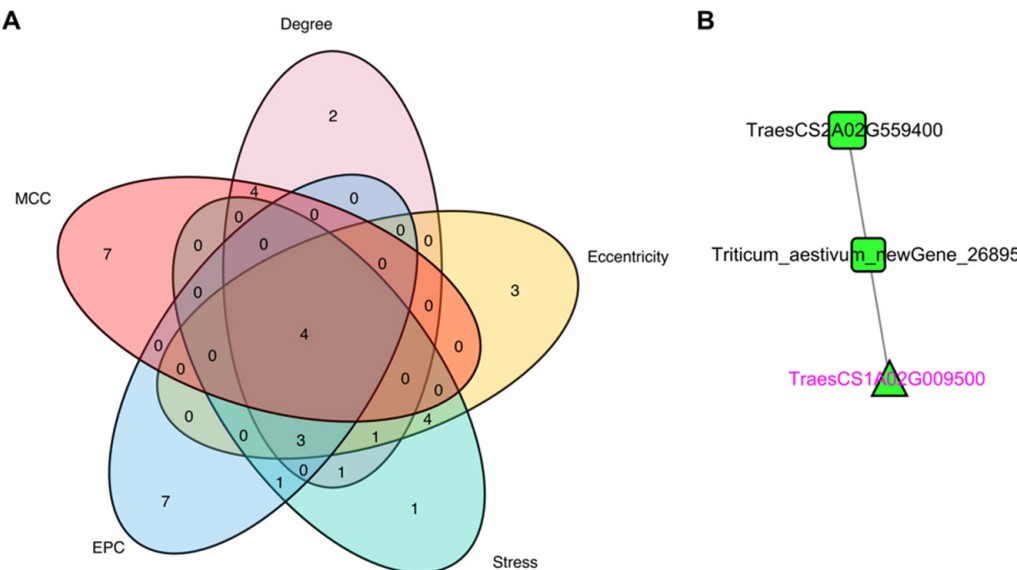

**Figure 6.** Hub genes and Hub modules: (**A**). Venn analysis showed that the four hub genes listed in Table 1 could be repeatedly screened by five approaches from cytoHubba with different analysis methods (Degree, EPC, EcCentricity, Stress, and MCC). (**B**). The screened hub PPI module 5 by MCODE. Note: the triangles indicate the seed DEGs. The complete results are shown in Figure S6. EPC, Edge Percolated Component; MCC, Maximal Clique Centrality; PPI, Protein–Protein Interaction; MCODE, Molecular COmplex Detection; DEGs, Differentially Expressed Genes.

**Table 1.** The common hub DEGs in the PPI network screened by centrality methods.

| #ID | CK vs. SP | Swiss_Prot_Annotation |
| --- | --- | --- |
| *Triticum_aestivum_newGene_7669* | down | Transcription factor HY5 OS = Solanum lycopersicum OX = 4081 GN = HY5 PE = 2 SV = 1 |
| *Triticum_aestivum_newGene_26895* | up | – |
| *TraesCS1A02G009500* | up | Receptor kinase-like protein Xa21 OS = Oryza sativa subsp. japonica OX = 39947 GN = XA21 PE = 1 SV = 1 |
| *TraesCS2A02G559400* | up | Probable LRR receptor-like serine/threonine-protein kinase At3g47570 OS = Arabidopsis thaliana OX = 3702 GN = At3g47570 PE = 2 SV = 1 |

*3.6. DNA MethylRAD Sequencing and DNA Methylation Levels*

The DNA methylation at the CCGG and CCWGG (W = T or A) sites in the entire genomes of SP and CK groups was examined by methylRAD sequencing. After data filtering, a total of 140.69–177.60 million raw reads were obtained for the six libraries, with 25.36–33.96 million clean reads produced, 95.01–98.17% of which were mapped to the reference wheat genome and 8.08–8.48% of which were uniquely mapped reads (Table S6), with sequencing analysis results that were consistent with those of previous studies [43]. The raw reads were uploaded to NCBI (SRA accession no. PRJNA827455). As shown in Figure 7A and Table S7, while the DNA methylation level at CCGG sites was higher than that at CCWGG sites in all samples, more sites of both types were detected in the SP group

than in the CK group, i.e., 7.98% and 4.01% for CCGG and CCWGG sites, respectively, indicating that methylation changes may have influenced ZKNM1 leaf spot formation.

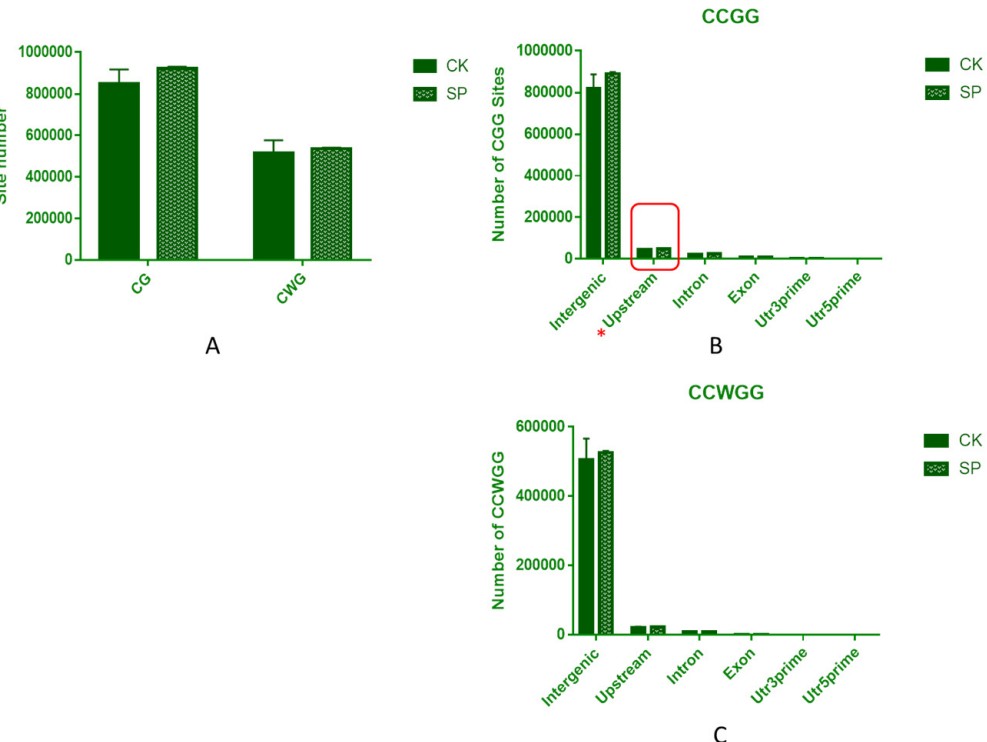

**Figure 7.** The methylated sites in the CG and CWG contexts: (**A**). Summary of methylated sites in the SP and CK groups. (**B**,**C**). Distribution of CCGG (**B**) and CCWGG (**C**) methylation sites within genome functional elements. Note: The "*" in the figure indicates a significant difference between the CK and SP groups in this element at the *p* < 0.05 level.

*3.7. Distribution of Methylation Sites within Different Functional Elements and Gene Regions*

For CCGG and CCWGG sites, the distribution patterns of methylated sites were compared at different elements of the genome. Both CCGG and CCWGG methylated sites were identified in intergenic regions, followed by upstream (2000 bp upstream of the transcription start site, TSS) and intron regions (Figure 7B,C). However, in the CCGG type, significantly more methylated sites were discovered only in the upstream regions in the SP group (47,855) than in the CK group (43,928), indicating that variation in methylation levels of promoters may play an important role in inducing leaf color variation.

The methylation site distribution was analyzed within the transcription start site (TSS), transcription termination site (TTS), and gene body regions (Figure S7). The methylated sites of the TSS were more frequent than those of the TTS and gene body regions. The difference in methylation levels at the TSS between the SP and CK groups was examined further (Figure 8). Generally, higher methylation levels in the SP group were detected upstream of the TSS (the most significant difference at approximately −1690 bp). However, considerably lower methylation levels were observed in the SP group near the beginning of the gene body, particularly 100 bp downstream of the transcription start site.

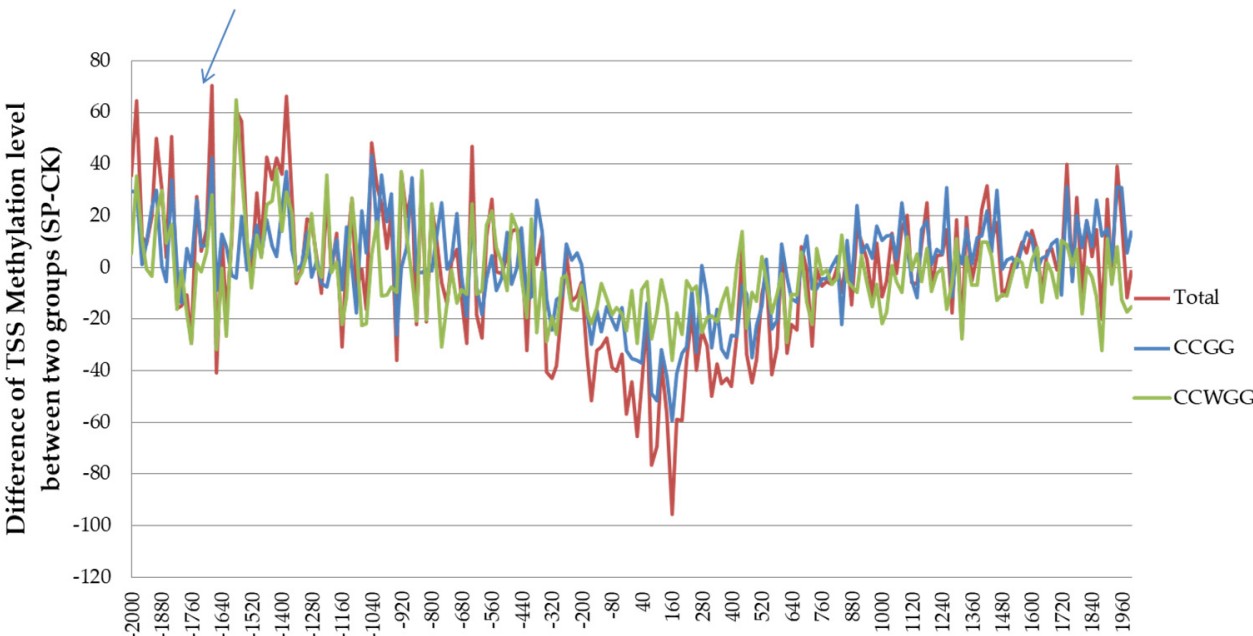

**Figure 8.** Differences in DNA methylation levels at the TSS region between the SP and CK groups. Note: The arrow indicates the highest difference in methylation levels between SP and CK groups at about −1690 bp.

### 3.8. The Differentially Methylated Sites and Functional Analysis

The differentially methylated sites (DMSs) in the CK and SP groups were used to further discover methylation differences between leaves with and without greenish spots. A total of 7860 CCGG and 5550 CCWGG sites were found to be DMSs (Figure 9A; Additional file 5). There were 3916 and 3265 hypermethylated CCGG and CCWGG sites, respectively, as well as 3944 and 2285 hypomethylated CCGG and CCWGG sites, respectively. A total of 17,946 genes were assigned by DMSs, only 721 of which harbored differentially methylated sites in the promoter region and thus were designated as DMPs for the further GO and KEGG enrichment analysis. A total of 38 highly enriched GO terms were assigned to these DMPs, as illustrated in Figure S8A and Additional file 6. The biological process category's main terms included "metabolic process" and "cellular process". The "cell", "cell part", and "membrane" terms were the three primary cellular components. Moreover, molecular functions such as "binding" and "catalytic activity" were strongly represented. According to the KEGG enrichment analysis (Figure S8B; Additional file 7), 22 DMPs were significantly enriched ($p$ value < 0.05) in 8 pathways, including "Glycosaminoglycan degradation", "Other glycan degradation", "Glycosphingolipid biosynthesis-ganglio series", "Brassinosteroid biosynthesis", "Various types of N-glycan biosynthesis", "Aminoacyl-tRNA biosynthesis", "Glycosphingolipid biosynthesis-globo and isoglobo series", and "Cutin, suberine, and wax biosynthesis".

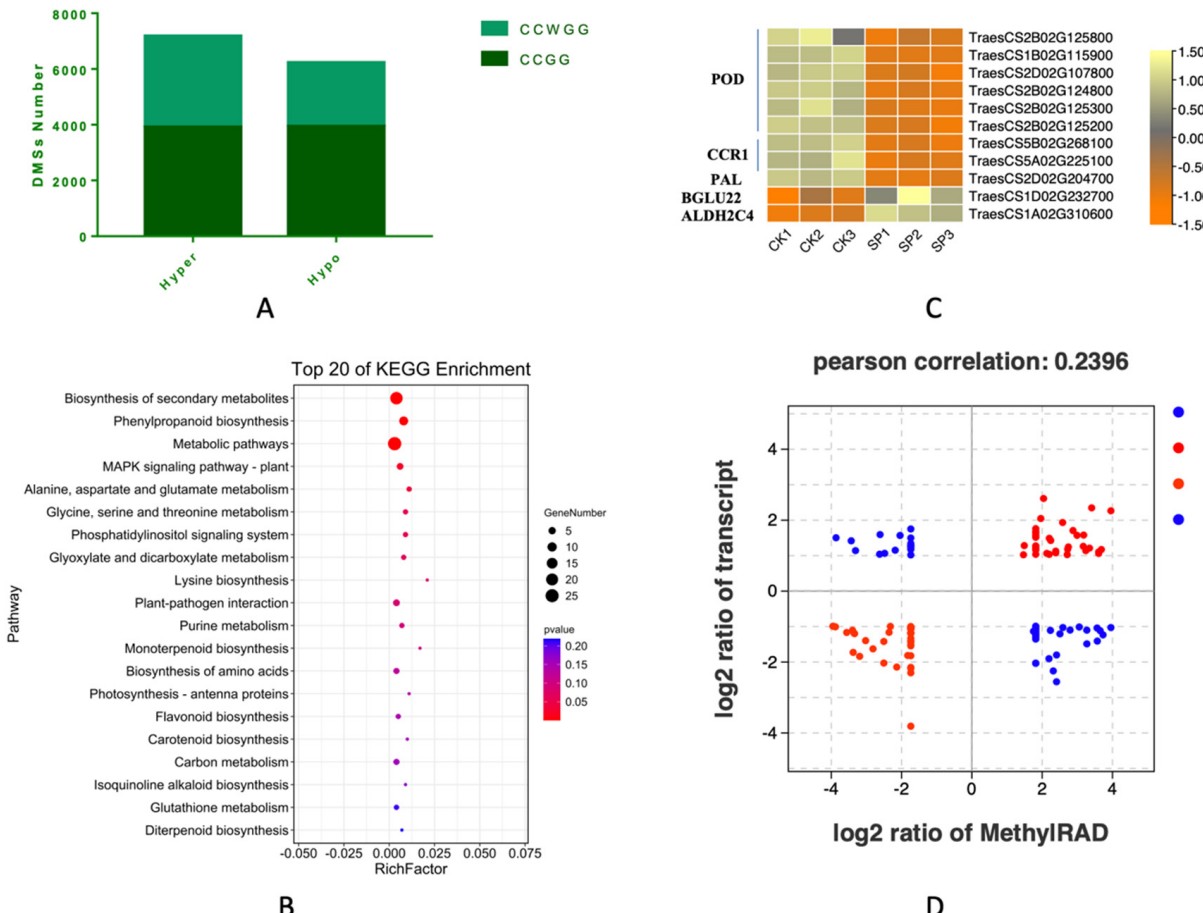

**Figure 9.** The integrated analysis of DMSs and DEGs: (**A**). The numbers of DMSs at the CCGG and CCWGG sites. (**B**). KEGG enrichment of the common genes differentially expressed and differentially methylated. (**C**). The expression profiles of the methylation regulated DEGs annotated in the "phenylpropanoid biosynthesis" pathway. The relative expression values ranged from −1.5 to 1.5. (**D**). Four-quadrant analysis between DMSs and DEGs. Blue spots represent the genes that are negatively correlated with their DNA methylation levels, while red spots represent genes with a positive correlation. DMSs, Differentially Methylated Sites; KEGG, Kyoto Encyclopedia of Genes and Genome; DEGs, Differentially Expressed Genes.

### 3.9. Integrated Analysis of DMSs and DEGs

A total of 131 DEGs listed in Table S8 were also screened by methylation analysis in our study. Ten were methylated in upstream regions that may be considered promoters, defining them as DMPs. The 131 DEGs were then assigned to KEGG pathways (Figure 9B). The three most represented KEGG pathways were "biosynthesis of secondary metabolites", "phenylpropanoid biosynthesis", and "metabolic pathways". Among 11 methylated DEGs enriched in phenylpropanoid biosynthesis (Figure 9C), 6 were PODs, 2 were CRR1s, and 1 was a PAL, related to defense. All of them were significantly downregulated, two upregulated DEGs were BGLU22 and ALDH2C4.

A four-quadrant graph (Figure 9D) was created to better examine the possible association between DNA methylation and gene expression, with 16, 38, 36, and 36 genes positioned in the first to fourth quadrants, respectively. The hub gene identified by the DEG PPI network, TraesCS1A02G009500, was discovered in the first quadrant. This hypomethylated gene interacted with the other five DEGs exhibiting methylation level variation (Figures 5 and 6), suggesting that this network is hypersensitive to DNA methylation regulation. Interestingly, one of the five dynamically methylated DEGs linked to TraesCS1A02G009500,

TraesCS6A02G363100, an unknown protein gene, was hypermethylated at −1689 bp in the SP group. This gene was found in the fourth quadrant.

In addition, the correlation analysis between DEGs and DMPs discovered nearly 3,223,147 pairs of DEG-DMP have a correlation coefficient >0.80 at the significant level of <0.05 ($p < 0.05$) (Additional file 8), demonstrating the tight relationship between the methylation level and gene expression. Among them, the methylation level of TraesCS6A02G363100 at −1689 bp of TSS was significantly negatively correlated with the expression of TraesCS1A 02G009500 (−0.93).

## 4. Discussion

In addition to traditional DEG analysis based on the total RNA sequencing data, GSEA has also become a common method to analyze gene expression data derived from a predefined gene set, which can detect modest but coordinated expression of functionally related gene groups [35]. In this study, both GSEA and traditional DEG analysis primarily highlighted the chlorophyll metabolism-related genes that could explain the spot development in ZKNM1. Among them, the downregulated and upregulated genes (*HEMA1s* and *PORAs*) involved in chlorophyll biosynthesis, respectively, might comprise a double block, suppressing chlorophyll levels, and thus they might be the major leaf spot inducers in ZKNM1. Therefore, it can be speculated that the loss of chlorophyll accumulation leads to leaf spot emergence.

Many vital genes have three homeologous copies in allohexaploid wheat, and the homeologous compensation and gene dosage effect reflect its hexaploid nature [44,45]. In this study, six transcripts (two on 1A, two on 1B, and two on 1D) were annotated as *HEMA1s*, encoding the first limiting enzyme beginning chlorophyll synthesis [10], and were all significantly downregulated. There have been few reports about wheat *HEMA1s* and their regulators. McCormac and Terry found that ELONGATED HYPOCOTYL 5 (HY5), a photoreceptor and central repressor in the light signaling pathway that promotes photomorphogenesis, plays a vital role in regulating the transcription levels of *Arabidopsis HEMA1* [46]. In this study, only one DEG was annotated to HY5, *Triticum_aestivum_newGene_7669*. This *HY5* DEG was a new putative gene in ZKNM1 with considerably lower expression in the SP group, which is consistent with the changing trends of *HEMA1s*. Thus, we deduced that HY5 in ZKNM1 might mediate *HEMA1s* and chlorophyll generation in SP groups (Figure S9).

According to PPI network analysis, *Triticum aestivum newGene 7669*, the HY5 gene, was a hub gene with the highest degree in the module primarily engaged in chlorophyll metabolism (Figures 5 and 6; Table S5) and linked to the other 29 DEGs as a potential transcription factor, the majority of which were possibly controlled by it. *TraesCS1A02G009500*, encoding a receptor kinase-like protein, was another hub gene and seed gene in this module. In rice, a homologous gene termed *Xa21* provides resistance to many pathogenic diseases [47]. This interaction between the annotated HY5 (encoded by *Triticum aestivum newGene 7669*) and TaXa21 (encoded by *TraesCS1A02G009500*) has been further validated by Y2H assay (Figure 10), indicating a possible association between the formation of leaf spots and the resistance response (Figure S9). Indeed, multiple investigations have discovered leaf spot mutants that provide increased plant disease resistance to a broad spectrum of diseases [5,40,48,49]. These spot mutants typically exhibit necrotic lesions and hypersensitive response processes, such as the accumulation of reactive oxygen species (ROS) and activation of pathogenesis-related genes [5,40], and are hence referred to as lesion mimics mutants (LMMs). The spots in ZKNM1 can, however, fade and turn green on their own. As a result, the control and production processes of the green and recoverable spots in ZKNM1 are complex and require further exploration.

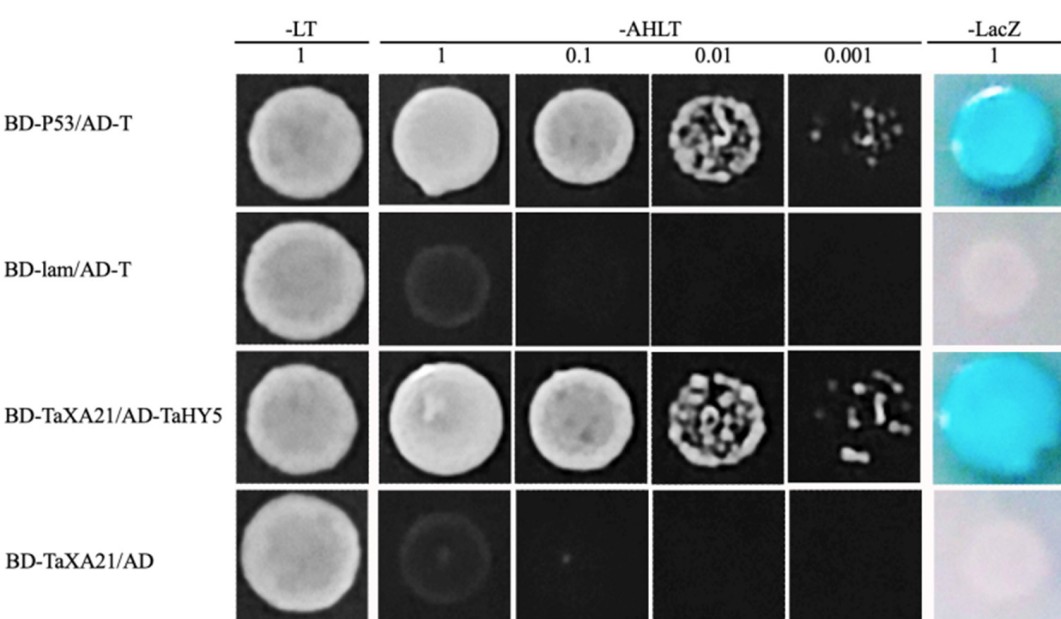

**Figure 10.** The validated interaction between TaHY5 (encoded by *Triticum aestivum newGene 7669*) and TaXa21 (encoded by *TraesCS1A02G009500*).

Previous research has demonstrated the tissue specificity of methylation signatures, resulting in heritable phenotypic differences [50,51], such as pigmentation [52,53]. The inherent spots in ZKNM1 could also be affected by methylation. According to previous research, CG context methylation is the most common kind of cytosine methylation [54]. Methylation within gene promoters had the most significant methylation ratio among genic areas [19]. This work discovered only 10 DEGs with substantially different methylation levels in promoters (DMPs), which may play a particular role in the production of ZKNM1 leaf spots.

The most distinct location for differences in methylation in the SP group was near the 1690 bp upstream of the TSS, which might be the more frequently methylated site in SP groups. *TraesCS6A02G363100*, hypermethylated at −1689 bp in the SP group was found in the fourth quadrant, indicating that the hyper-methylation level in the promoter may cause its suppressed transcription level, which may be involved in leaf spots formation in ZKNM1, linking both transcriptional and methylation regulation. This finding is consistent with previous research, which demonstrated that promoter hypermethylation inhibits regulatory protein binding and represses transcription [19]. The relationship between this unknown protein encoded by *TraesCS6A02G363100* and its linked TraesCS1A02G009500 protein (homologous to Xa21), which is connected to HY5, suggests that methylation regulation may mediate chlorophyll content (Figure S9). The significant correlation of the methylation level of *TraesCS6A02G363100* with the expression of *TraesCS1A02G009500* and *Triticum aestivum newGene 7669* (Additional file 8) to some extent supported this deduction, which nevertheless requires additional validation.

This work discovered that chlorophyll metabolism was primarily responsible for producing greenish leaf spots in a high-yield wheat variety ZKNM1.

## 5. Conclusions

The current study evaluated the role of transcriptional control by DNA methylation on leaf color variation in the wheat variety ZKNM1. The methylation site at −1690 bp upstream of the TSS with the highest difference in methylation levels was highlighted, and ten DEGs with variation in their methylation level at promoter regions were finally identified, one of which (*TraesCS6A02G363100*) possibly interacted with a protein kinase gene *TraesCS1A02G009500*, which was hypermethylated at −1689 bp of the TSS possibly inhibiting transcription. Furthermore, two types of chlorophyll biosynthesis limiting genes,

*HEMA1s* and *PORAs*, were discovered to be considerably downregulated in ZKNM1 leaf spot tissues, suggesting that the decreased expression of chlorophyll genes may drive spot formation. Further investigation suggested that the transcription factor HY5, encoded by a putative novel gene in ZKNM1 (*Triticum aestivum newGene 7669*), was likewise related to the hub gene *TraesCS1A02G009500* and controlled *HEMA1s* expression. Furthermore, this connection might be attributed to methylation modulation. These highlighted pathways and hub genes putatively governing leaf spots in ZKNM1 might provide insights into improving photosynthesis.

**Supplementary Materials:** The following are available online at https://www.mdpi.com/article/10.3390/agronomy12071519/s1, Figure S1: The correlation analysis (A) and PCA (B) of six samples, Note: No1-0-1,2,3 represents CK1, 2, 3 and No1-1-1,2,3 represents SP1, 2, 3, respectively. Figure S2: The DEG analysis between CK and SP groups, Note: A. Volcano plot between two types of leaf samples. B. GO enrichment of DEGs. Figure S3: The directed acyclic graphs for GO enrichment of DEGs, Figure S4: qRT-PCR validation, Figure S5: The expression profiling of genes involved in carotenoid (A) and anthocyanin (B) biosynthesis, Note: The relative expression values range from −2 to 2. Figure S6: The PPI hub network modules. Note: The triangles indicate the seed DEG. Figure S7: DNA methylation levels in different gene regions, Note: NM1-0-1,2,3 represents CK1, 2, 3 and NM1-1-1,2,3 represents SP1, 2, 3, respectively. Figure S8: GO (A) and KEGG (B) enrichment of DMPs, Figure S9: The possible regulators of the leaf spot formation in ZKNM1. Note: The circles stand for the interaction proteins encoded by the DEGs based on PPI analysis. Rectangles represent protein-regulated chlorophyll synthesis-related genes. The fill color is yellow for transcripts that are down-regulated and green for transcripts that are up-regulated. The font red represents up-regulated methylation levels, white represents down-regulated methylation levels, and black represents unchanged methylation levels, Table S1: The primers for qPCR, Table S2: The transcriptome sequencing data obtained in this study, Table S3: The annotated new gene statistics, Table S4: Gene set enrichment analysis (GSEA) between CK and SP groups, Table S5: Top 10% genes evaluated in the protein–protein interaction (PPI) network, Table S6: The mapping statistics of all reads from the MethylRAD libraries; Table S7: Summary of methylation site coverage; Table S8: The 131 DEGs with significantly different methylation level between CK and SP groups.

**Author Contributions:** Conceptualization, X.F. and T.W.; methodology, Z.X.; software, X.F.; validation, Q.Y. and T.W.; formal analysis, X.F.; investigation, Z.X., B.F. and Q.Z.; resources, T.W.; data curation, X.F. and F.W.; writing—original draft preparation, X.F. and Z.X.; writing—review and editing, F.W.; visualization, F.W.; supervision, Q.Y. and T.W.; project administration, Q.Y. and T.W.; funding acquisition, X.F. All authors have read and agreed to the published version of the manuscript.

**Funding:** This research was funded by Key R&D Projects of Sichuan Province, grant No. 2019YFN0024 and Youth Innovation Promotion Association of CAS, grant No. 2020364.

**Institutional Review Board Statement:** Not applicable.

**Informed Consent Statement:** Not applicable.

**Data Availability Statement:** The RNA-Seq and MethylRAD raw reads were uploaded to NCBI (SRA accession no. PRJNA827248 and PRJNA827455).

**Acknowledgments:** We thank Shanghai OE Biotech. Co., Ltd. (Shanghai, China) and Biomarker Technologies Co., Ltd. (Beijing, China) for their high throughput sequencing service and bioinformatics support.

**Conflicts of Interest:** The authors declare no conflict of interest.

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
