# Peer review of "DNA Methylation and RNA-Sequencing Analysis to Identify Genes Related to Spontaneous Leaf Spots in a Wheat Variety ‘Zhongkenuomai No.1’"

_agronomy, doi:10.3390/agronomy12071519_

Round 1

Reviewer 1 Report

The Authors are resubmitting their manuscript following my advise not to consider the leaf spot trait as of ornamental interest. I think that the paper can be published, after considering the following.

Line 20

“ a methylation hot site was discovered 1690 bp upstream of the transcriptional start site…” is not clear. I would suggest: “genome-wide methylation analysis indicates a that a hypermethylated region is present  1690 bp upstream of the transcriptional start sites…”

Line 25

 “gene from in interaction”: this is not correct; remove “from”?

Immune response: this concept appears abruptly and the connection with photosynthesis here is not understandable. Either introduce it or remove.

Line 246

“Three replicate correlation coefficients”: do you mean “the correlations between replicates”?

Line 470

“Three copies among the three subgenomes” replace with “three homeologous copies “

Line 476

“which might have been caused by the variation in their upstream genes encoding common regulators rather than the DEGs themselves”: hypothetical and obvious, delete.

Line 517

“Furthermore, which location influences gene regulation is always noteworthy”: Delete or rephrase more clearly.

Line 538

“and thus inhibited transcription”. Replace with “possibly inhibiting

Author Response

Response to Reviewer 1 Comments

The Authors are resubmitting their manuscript following my advise not to consider the leaf spot trait as of ornamental interest. I think that the paper can be published, after considering the following.

Point 1: Line 20: “a methylation hot site was discovered 1690 bp upstream of the transcriptional start site…” is not clear. I would suggest: “genome-wide methylation analysis indicates a that a hypermethylated region is present 1690 bp upstream of the transcriptional start sites…”

Response 1: Thanks for your suggestions. We have corrected it in the revised manuscript.

Point 2: Line 25: “gene from in interaction”: this is not correct; remove “from”? Immune response: this concept appears abruptly and the connection with photosynthesis here is not understandable. Either introduce it or remove.

Response 2: Thanks for your suggestions. We have corrected it in the revised manuscript.

Point 3: Line 246: “Three replicate correlation coefficients”: do you mean “The correlations between replicates”?

Response 3: Thanks for your suggestions. We have corrected it in the revised manuscript.

Point 4: Line 470: “Three copies among the three subgenomes” replace with “three homeologous copies”

Response 4: Thanks for your suggestions. We have corrected it in the revised manuscript.

Point 5: Line 476: “which might have been caused by the variation in their upstream genes encoding common regulators rather than the DEGs themselves”: hypothetical and obvious, delete.

Response 5: Thanks for your suggestions. We have corrected it in the revised manuscript.

Point 6: Line 517: “Furthermore, which location influences gene regulation is always noteworthy”: Delete or rephrase more clearly.

Response 6: Thanks for your suggestions. We have corrected it in the revised manuscript.

Point 7: Line 538: “and thus inhibited transcription”. Replace with “possibly inhibiting transcription”.

Response 7: Thanks for your suggestions. We have corrected it in the revised manuscript.

Reviewer 2 Report

Comments.

1)    Avoid acronyms in the abstract, if used it must be defined/spelled out.

2)    The writing needs improvement, needs to be reviewed by an English language expert.

3)    Line 127: what do you mean by perfect match? Do you mean uniquely mapped reads? Please report the reads mapped, reads unmapped, uniquely mapped reads, mismatches and so on for all samples as a supplementary data

4)    Instead of p-value, q-value (FDR) should be used for significant results (for DEGs and DMSs)

5)    Line 135. Fold change should be >2. Why it is written it as <0.5? Did you used 2 or 0.5. please clarify and correct all the results accordingly.

6)    Did author pool the DNA and RNA from same sample? Did you crush the sample together and separated for DNA and RNA? Please clarify and mention in paper.

7)    Methylation data was mapped using bowtie2, then how about methylation calling? Which software was used? Please mention in material and method section.

8)    As, DNA methylation regulates gene expression. Integrative analysis of DNA methylation (DMSs) and gene expression (DEGs) must be performed. It will generate some interesting results to discuss.

Author Response

Response to Reviewer 2 Comments

Point 1: Avoid acronyms in the abstract, if used it must be defined/spelled out.

Response 1: Thanks for your suggestions. We have corrected it in the revised manuscript.

Point 2: The writing needs improvement and needs to be reviewed by an English language expert.

Response 2: Thank you very much. We have revised the language.

Point 3: Line 127: what do you mean by perfect match? Do you mean uniquely mapped reads? Please report the reads mapped, reads unmapped, uniquely mapped reads, mismatches and so on for all samples as a supplementary data.

Response 3: Thank you very much for your advice. We have modified this sentence (Line 193-194) and provided the relative results about sample sequencing in supplementary Table S2.

Point 4: Instead of p-value, q-value (FDR) should be used for significant results (for DEGs and DMSs)

Response 4: Thanks for your suggestion. In our study, Q value (FDR) was used for RNA-seq analysis in this study. Line 206-208: The resulting Pvalue was adjusted using Benjamini and Hochberg’s approach for controlling the false discovery rate (FDR). Since It’s easy to misunderstand by using “corrected P value” to represent “Q value (FDR)” before, we have corrected it to “adjusted P value (Q value, FDR)” in the revised version.

Besides, P value was used for methylation analysis in this study. We completely agree with your valuable suggestion that Q value (FDR) should be used for methylation analysis. The Q value is an adjusted P value, taking in to account the false discovery rate (FDR). A P value of 0.05 implies that we are willing to accept that 5% of all tests will be false positives. An adjusted P value of 0.05 implies that we are willing to accept that 5% of the tests found to be statistically significant will be false positives. Such an adjustment is necessary when we’re making multiple tests on the same sample. As a matter of fact, due to the hub genes were obtained by taking the common part of DEGs and DMSs and it will take a lot of time to reanalyze the data, the DMSs is limited to P value in this study. In the future study, we will adopt your suggestion.

Point 5: Line 135. Fold change should be >2. Why it is written it as <0.5? Did you used 2 or 0.5. please clarify and correct all the results accordingly.

Response 5: Thanks for your suggestions. In fact, fold change > 2 was considered as significantly up-regulated, while fold change < 0.5 was considered significantly down-regulated. Both up-regulated and down-regulated were differentially expressed genes. For accurate description, we changed it to “Adjusted P value (Q value, FDR) < 0.05 & |log2 fold change|>1 was set as the threshold for significantly differential expression” (Line 208-209).

Point 6: Did author pool the DNA and RNA from same sample? Did you crush the sample together and separated for DNA and RNA? Please clarify and mention in paper.

Response 6: Thanks for your suggestions. In our study, each replicate was crushed together and separated for DNA and RNA to ensure them pooled from the same sample. We have modified the methods sections in the revised manuscript (Line 174-175).

Point 7: Methylation data was mapped using bowtie2, then how about methylation calling? Which software was used? Please mention in material and method section.

Response 7: Thank you very much. MethylRAD analysis does not involve calling, the FspEI enzyme recognizes CG sites that are methylated for digestion, as long as the cut CG sites are methylated, so the methylation level of the site can be obtained by counting the depth of each site after bowtie comparison with the comparison software, see the flowing technical article for details (Wang et al., 2015). Wang S#, Lv J#, Zhang L#, Dou J, Sun Y, Li X, Fu X, Dou H, Mao J, Hu X & Bao Z*. (2015) MethylRAD: a simple and scalable method for genome-wide DNA methylation profiling using methylation-dependent restriction enzymes. Open Biology. 5: 150130.

Point 8: As, DNA methylation regulates gene expression. Integrative analysis of DNA methylation (DMSs) and gene expression (DEGs) must be performed. It will generate some interesting results to discuss.

Response 8: Thanks very much for your advice. We have added more integrative analysis between gene expression and methylation and presented them in Additional 8 as well as the 3.9 section (Line 570-575) and 4 discussion section (Line 667-670).

This manuscript is a resubmission of an earlier submission. The following is a list of the peer review reports and author responses from that submission.

Round 1

Reviewer 1 Report

After reading this re-submitted manuscript, I remain very doubtful about the association of ornamental crops with indoor/vertical farming.
To be useful, ornamental crops must be visible to a large number of people, while indoor/vertical farms would probably not be open to the public, for sanitary and management reasons. Also, I am not convinced of the ornamental relevance of this particular trait of wheat, transient leaf spots/stripes.

Therefore, I think that this work can be interesting as a study of a leaf spot trait, and can provide useful information on a chlorophyll biosynthesis mutant, but is not well placed in a special issue on “Role of Vertical Farming in Modern Horticultural Crop”.
The title assumes that this wheat variety is already used  as an ornamental plant, but this is not the case, so the title needs to be changed anyway.

Moreover, the Zhang et al. 2018 paper (Line 55) does not deal with ornamental use of wild barley, it just mentions that it can be planted as an ornamental. It is not a good backup for the ornamental use of wheat.

Consequently, the statement that justifies this study: “In this study, we conducted a transcriptome and DNA methylation investigation on ZKNM1 to identify the related pathways or genes associated with the leaf spot dynamics and use them to facilitate future indoor ornamental wheat breeding”, in my opinion, should not include the last sentence, and the paper may be published in Agronomy, rather than in a special issue on vertical farming.

However, the responsibility of deciding if the manuscript fits in the topic of the special issue is the Editors’.

Coming to the quality of the manuscript, it was significantly improved with respect to the first submission, but some improvements are still necessary, in my opinion.

Materials and methods. Leaf sampling is still not adequately described. How many plants? Only three (three flag leaves)? How much tissue/leaf punches were taken and used per DNA/RNA extractions? The number is important for representativeness of the sample.

125: “variance” is not the right word here, I would write “trait”.

137: figure 1 does not show “performance” of Zhongkenuomai

156: “those containing only one exon were excluded”: this requires a justification and an explanation.

243: “to identify the critical genes and their pathways triggering leaf spot development.” This study cannot accomplish such identification, but only identifying genes that are differentially expressed.

276: ES should be written in extenso.

279: NES should be written in extenso.

292: explain figure in caption

354: I think that this figure should be reproduced much larger than this, otherwise it will not be readable.

364: EPC, MCC should be written in extenso in caption.

381: “comparable” or compared?

444: Fig. 9 B and D area hardly readable.

445: “methylated” or differentially methylated?

459-462. I do not understand this discussion about carotenoid and anthocyanins. I would remove it.

471- 2: “in their common upstream regulators rather than the transcripts themselves” meaning “in their promoters”?

Discussion: it may be shortened  by avoiding to repeat results.

512: “which might be the easily methylated site in SP groups.”: I do not understand this

528-33: This part has little to do with the results of this work, in which no cause-effect relationship can be established between chlorophyll metabolism and leaf spots.

Additional files 1-7: I did not find these files in the Supplementary files folder.

Table S8: caption mentions 131 DEGS, but the table has 66 lines

Supplementary figures have no captions, but captions are necessary.

Finally, in the attached edited manuscript I deleted some unnecessary sentences and highlighted in green a few language mistakes

Reviewer 2 Report

Methods are poorly explained, especially bioinformatics part.  Please re-write the manuscript. Submit all the sequencing data to NCBI

Author Response

Point 1: Methods are poorly explained, especially bioinformatics part. Please re-write the manuscript.

Response 1: Thank you very much for your kind suggestions. We have modified the methods sections and added more detailed information in the revised manuscript, including the replicate numbers, bioinformatics analysis process, tools and so on.

Point 2: Submit all the sequencing data to NCBI

Response 2: Besides, the raw reads of RNA-Seq and MethylRAD had been uploaded to the NCBI (SRA Accession No. PRJNA827248 and PRJNA827455).

Reviewer 3 Report

  1. I'm not very sure about the ornamental value of wheat with leaf color variations. Maybe sometimes they look not so attractive. For example, the plant in figure 1, in my opinion, looks not so good in ornamental value.
  2. The scientific part of this study looks valuable in enriching the knowledge, particularly in gene regulation mechanisms for leaf spot formation.
  3. The authors should point out the future perspective more specified and clear in the section of the conclusion about ornamental wheat breeding.
  4. Avoid using first-person writing throughout the manuscript.
  5. Give full names of all abbreviations in every table and figure that treat them as an independent.
  6. L13: with both edible and decorative value  - with the both edible and decorative value
  7. L18: be coming - becoming
  8. L84: exhibitdistinct - exhibit distinct
  9. L215: What do you mean about "genebody" or "gene body"?
  10. The authors could organize a conclusive graph to enrich the discussion systematically.

Author Response

Point 1: I'm not very sure about the ornamental value of wheat with leaf color variations. Maybe sometimes they look not so attractive. For example, the plant in figure 1, in my opinion, looks not so good in ornamental value.

Response 1: Thank you. We actually intend to bring more attention to the future potential of leaf color variation for improving horticultural traits in wheat through this article. We hope that this will enable us to preserve more germplasm resources (in previous abnormal leaf color in food crops was considered as an unfavorable variation and needed to be improved or eliminated), broaden the scope of wheat applications, and help farmers increase the economic value of wheat.

Point 2: The scientific part of this study looks valuable in enriching the knowledge, particularly in gene regulation mechanisms for leaf spot formation.

Response 2: Thank you for your approval.

Point 3: The authors should point out the future perspective more specified and clearer in the section of the conclusion about ornamental wheat breeding.

Response 3: Thank you for your suggestion. We have added our opinions about the need to breed or develop ornamental wheats.

Point 4: Avoid using first-person writing throughout the manuscript.

Response 4: Thanks for your suggestions. We have corrected it in the revised manuscript.

Point 5: Give full names of all abbreviations in every table and figure that treat them as an independent.

Response 5: Thanks for your suggestions. We have already supplemented the full names of all abbreviations in note part of every table and figure.

Point 6: L13: with both edible and decorative value with the both edible and decorative value

Response 6: Thanks for your suggestions. We have corrected it in the revised manuscript.

Point 7: L18: be coming - becoming

Response 7: Thanks for your suggestions. We have corrected it in the revised manuscript.

Point 8: L84: exhibitdistinct - exhibit distinct

Response 8: Thanks for your suggestions. We have corrected it in the revised manuscript.

Point 9: L215: What do you mean about "genebody" or "gene body"?

Response 9: Gene body is defined as the entire gene from the transcription start site to the end of the transcript, including exons and introns.

Point 10: The authors could organize a conclusive graph to enrich the discussion systematically.

Response 10: Thanks for your suggestions. We have added a conclusive graph as Figure S9.
